# Influence of Non-Uniform Bluntness on Aerodynamic Performance and Aerothermal Characteristics of Waverider

**Zhipeng Qu** [1], **Wanyu Wang** [1], **Houdi Xiao** [2], **Yao Xiao** [3,4,*], **Guangli Li** [3,4] **and Kai Cui** [3,4]

1   School of Mechanical and Electrical Engineering, Henan Institute of Science and Technology, Xinxiang 453003, China
2   Beijing Aerospace Technology Research Institute, Beijing 100074, China
3   State Key Laboratory of High Temperature Gas Dynamics, Institute of Mechanics, Chinese Academy of Sciences, Beijing 100190, China
4   School of Engineering Science, University of Chinese Academy of Sciences, Beijing 100049, China
*   Correspondence: xiaoyao@imech.ac.cn

**Abstract:** The waverider is widely used in hypersonic vehicles with its high aerodynamic performance, but due to the serious aerothermal environment, its sharp leading edge should be blunted. Circular blunt is one of the commonly used aerothermal characteristic protection methods. Circular blunt with larger diameter can reduce peak heat flux, but at the same time, it will lead to larger drag. The existing research shows that under the same blunt diameter in two-dimensions, the non-uniform blunt can reduce the peak heat flux by 20%, and the difference of drag is small. In this paper, the non-uniform blunt profile is applied to the three-dimensional waverider, and the influence of the non-uniform blunt profile on the aerothermal characteristic performance and aerodynamic performance of the waverider is studied, and the results are compared with those of circular blunt. The numerical simulation is used to compare and analyze the waverider under different angles of attack, flight altitudes, and Mach number. The results show that the peak heat flux of the waverider with non-uniform blunt reduces by about 17% compared with that with circular blunt under a small angle of attack range, Mach 2-10, and a flight altitude of 15–35 km. Meanwhile, when the blunt height/diameter is 20 mm, the aerodynamic performance difference between the two different blunt profiles does not exceed 3% within a 15 degrees angle of attack, Mach 2-10, and flight altitude of 15–35 km. The non-uniform blunt profile can be applied to the design of the three-dimensional waverider.

**Keywords:** waverider; non-uniform; hypersonic; computational fluid dynamics (CFD)

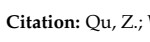



## 1. Introduction

When the aircraft flies at high speed, it will produce large shock drag and friction drag, resulting in the emergence of a "lift to drag barrier". Under the above condition, the traditional design idea of taking the wing as the main lifting component is rarely adopted, and the wing body integration design method is generally applied. The aerodynamic performance shapes of supersonic aircraft are generally divided into four categories: lift body, symmetrical body of revolution, blended wing body, and waverider. The waverider configuration is widely used in hypersonic vehicles with its good lift-to-drag ratio aerodynamic performance [1–8]. With the development of the research on hypersonic vehicles, the research on the waverider has gradually become one of the hotspots. References [1–3,7,8] carried out the design method for the waverider aircraft, References [4–6] have carried out research on the optimization of the waverider, and References [9–18] have studied the shape design of various waveriders and the aerodynamic shape design of hypersonic vehicles based on the waverider. The above studies have carried out a lot of research work on the waverider by numerical simulations, theory, and experiments, which showed that the waverider has a very broad application prospect.

The waverider is based on the principles of shock capture and streamline tracking, so the standard waverider is a sharp shape with no thickness at the leading edge. However, restricted by aerodynamic performance heating, the sharp leading edge needs to be blunted. The authors of [19] summarize the existing blunt methods of the waverider. The blunt methods of the waverider can be divided into three categories: the "adding material method" [20], the "reducing material method" [5], and the "mixing blunt method". Circular bluntness is a simple and effective method for the thermal protection, which is the most widely used method in engineering at present. Other profiles of the leading edge of the waverider are also used in the bluntness of the waverider. In [21–23], the power curves and artificial blunt front were used to blunt the waverider. The design of the above blunt methods is complex and not convenient for design and manufacture. Circular blunt is easy to design and manufacture, and widely used in the leading edge blunt of the waverider. Circular blunt will produce peak heat flow at the stagnation point, which is prone to point ablation.

As for the heat flux of the stagnation point, References [24–26] have proven that the peak heat flow is inversely proportional to the root of curvature radius at the stagnation point. Obviously, a greater blunt radius leads to a lower heat flux value at the stagnation point. However, a larger blunt radius leads to a higher drag. When using circular blunt to blunt the waverider, it is easy to produce point ablation in hypersonic flight. As the distance from the stagnation point increases, the heat flow gradually decreases. For circular blunt to produce point ablation at the stagnation point, to reduce the peak heat flow and make the heat flow near the stagnation point become uniformly distributed, an optimized 2D (non-uniform blunt) profile is obtained by resolving Navier–Stokes equations based on the genetic algorithm in [27]. The heat flux of the optimized 2D profile is distributed relatively uniform compared to that of the circular profile in the vicinity of the stagnation point. Compared to the circular leading edge, the peak heat flux of the optimized 2D profile is significantly reduced (Figure 1), but the pressure near the stagnation point of the optimized shape is significantly higher than that of the circular profile (Figure 2), which leads to higher drag.

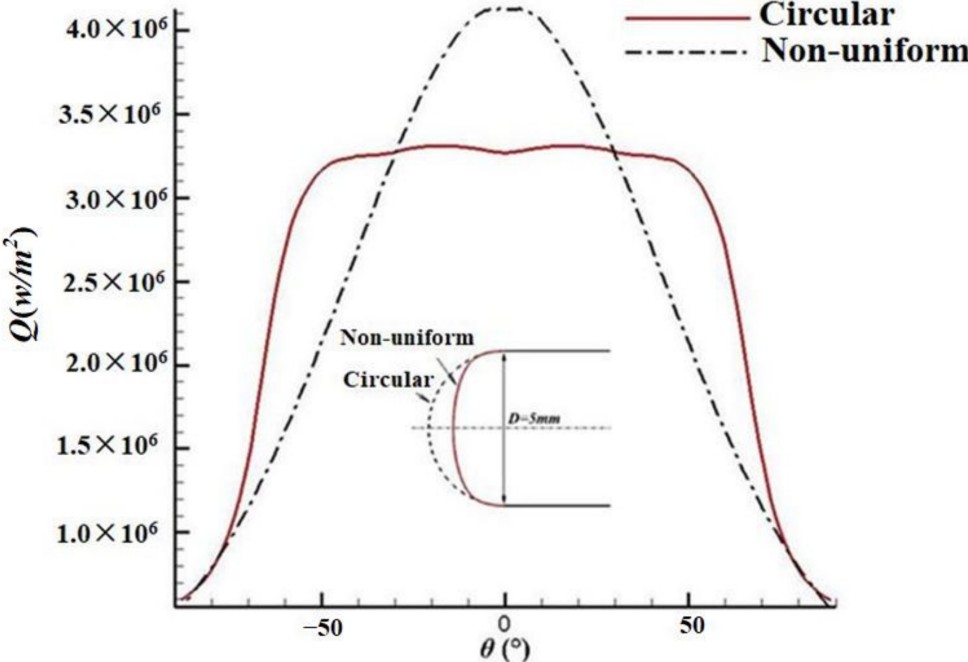

**Figure 1.** Heat flow distribution of circular and non-uniform leading edge [27].

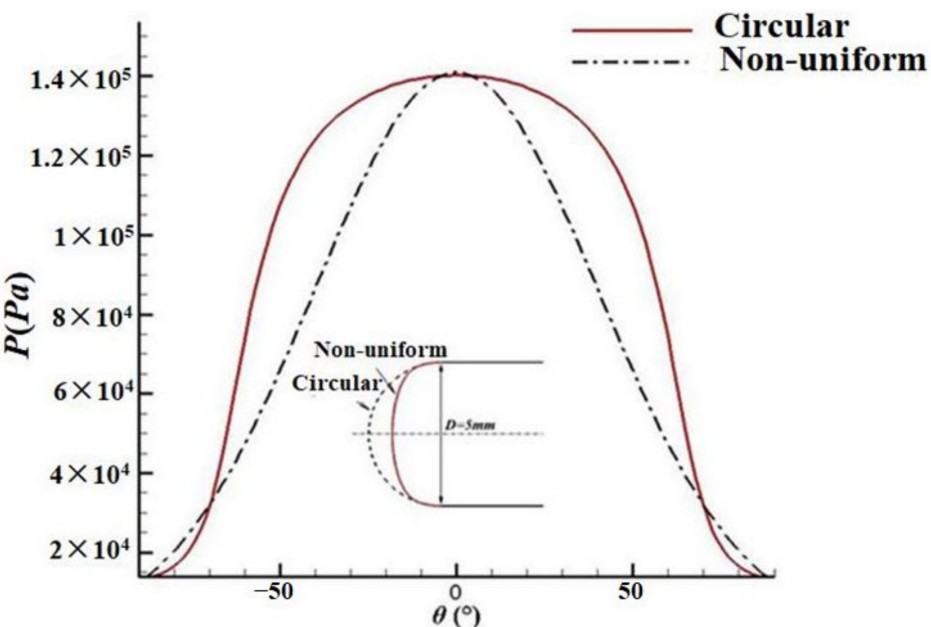

**Figure 2.** Pressure distribution of circular and optimal leading edge [27].

Based on the above research ideas, the two-dimensional non-uniform blunt profile is applied to the three-dimensional waverider vehicle to investigate its impact on the aerothermal characteristics and aerodynamic performance of the waverider and compare with the aerothermal characteristics and aerodynamic performance of circular blunt. In this paper, the purpose is to study the effects of the non-uniform blunt profile on the aerothermal characteristics and aerodynamic performance of the waverider. The influence of the circular shape and non-uniform blunt shape on the aerothermal characteristics and aerodynamic performance of the waverider is analyzed, and the performance of the two configurations under different angles of attack, flight altitudes, and Mach numbers is studied. The applicable range of the non-uniform blunt profile under different flight conditions is discussed.

## 2. Model Introduction

The waverider was generated based on an elliptical cone with an aspect ratio (width to length) of 0.618, the long axis of the elliptical cone angle was 7.09 degrees, and the length of the waverider was 2 m. The design Mach number was 6. The upper surface of the waverider is a plane. The numerical calculation of the waverider was carried out. The computational conditions are: the flight Mach number was 6 and the flight altitude was 25 km, the wall boundary condition was viscous adiabatic wall, and the turbulence model used the two-equation k-ε model. The pressure contour of the waverider under the 0-degree angle of attack is shown in Figure 3. The waverider was obtained in the inviscid flow field, and viscosity needs to be considered in flight, so there will be leakage at the edge of the waverider. It can be seen from Figure 3 that the lower surface of the waverider bears high pressure, and the upper surface has low pressure.

In the blunt method of the "adding material method", the upper surface of the waverider was raised to a certain height (the height is equal to the blunt diameter or blunt height), then the upper and lower edge surfaces of the waverider were blunted. The bluntness method and bluntness waverider are shown in Figure 4. With the increase of blunt radius/height, the volume of the waverider gradually increased.

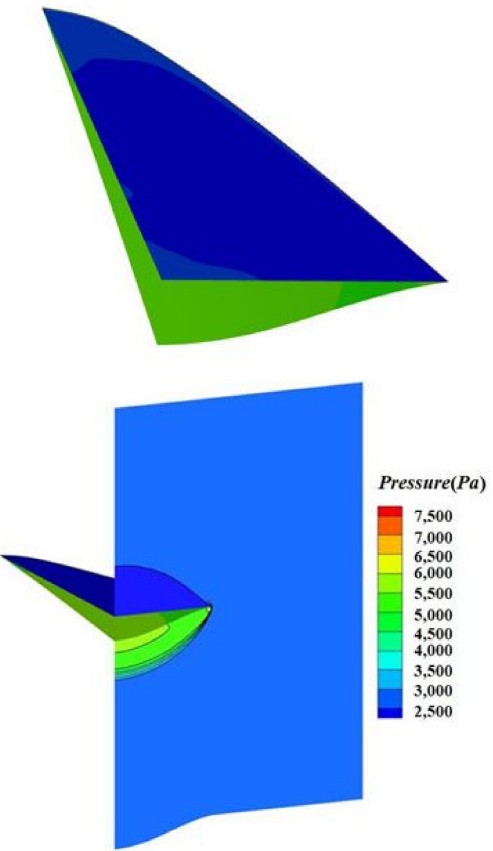

**Figure 3.** Pressure contour under the 0-degree angle of attack (units: Pa).

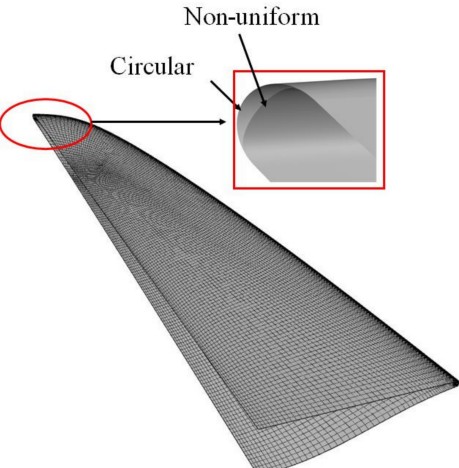

**Figure 4.** Bluntness methods and configuration of the waverider.

In this paper, the effects of two shapes (circular profile and non-uniform profile) on the aerothermal characteristics and aerodynamic performance force of the waverider are investigated, including angle of attack, flight altitude, and Mach number. The numerical simulation conditions are shown in Table 1. The datum marked after the number indicates that the waverider was under design conditions. The wall temperature of the waverider was different under different flight conditions. This paper mainly investigates the aerodynamic heat and aerodynamic performance of two kinds of blunt waverider. For the convenience of analysis, the wall temperature was set as 300 K.

**Table 1.** Layout of numerical plan.

| Type | Value | Blunt Diameter/Height |
|---|---|---|
| Angle of attack (degrees) | 0 (base), 5, 10, 15, 20 | |
| Flight altitude (km) | 15, 20, 25 (base), 30, 35 | 4, 10, 20 |
| Mach number | 2, 4, 6 (base), 8, 10 | |

For the convenience of calculation, half of the waverider was used for analysis and calculation. The length of the waverider was 2 m. The grid of the waverider is shown in Figure 5.

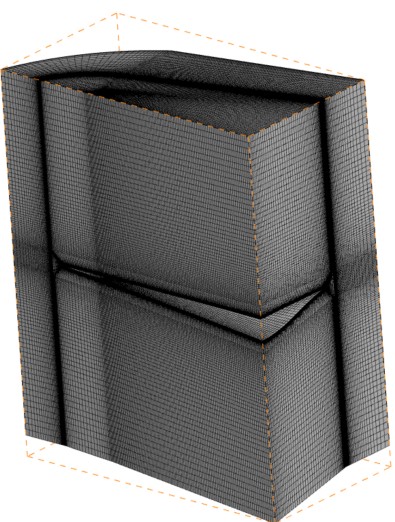

**Figure 5.** The grid of the waverider.

## 3. Numerical Method Validation and Grid Convergence Tests

In order to obtain accurate calculation results, it is necessary to verify the aerothermal characteristics and the aerodynamic performance force model. The validity of aerothermal characteristics and aerodynamic performance models was verified by comparing the calculation results of the simulation model and the standard model (test).

The research group has carried out aerodynamic performance model verification, and in order to avoid repetition, we will not repeat it again. The computational conditions, model validity verification, and grid convergence tests are the same as those in [19], and more details can be found there.

The validity of the aerothermal characteristic model was verified by the straight biconic test results and numerical simulation results. The straight biconic model [28] was used to validate the reliability of the solver code and is shown in Figure 6. The numerical simulation computational conditions were: the Mach number was 9.86, the Reynolds number was 231,800, the incoming flow temperature was 48.88 k, the wall temperature was 300 K, and the turbulence model was the k-$\varepsilon$-Rt model. The grid number was 688,000, and the first near-wall size was L $\times 10^{-5}$ (where L is the length of the straight biconic model).

A comparison between blunt biconical numerical results and NASA results is shown in Figure 7. The abscissa in the figure is the normalized coordinate and the ordinate is the heat flow value. In the figure, "numerical" represents the calculation results of the code used in this paper and "NASA" represents the NASA results. It can be seen from the figure that the numerical results are in good agreement, indicating that the code solver has high reliability.

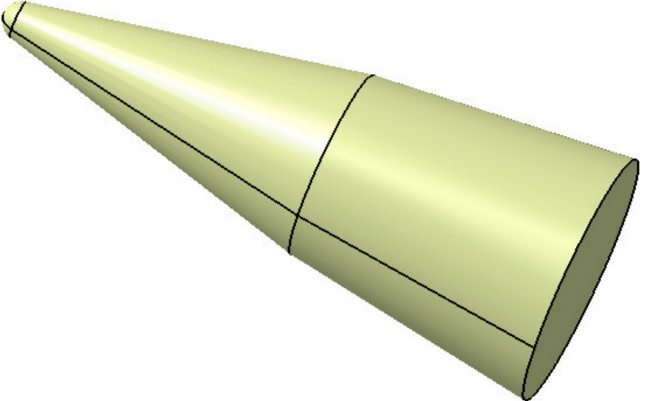

**Figure 6.** Straight biconic model (units: mm) [28].

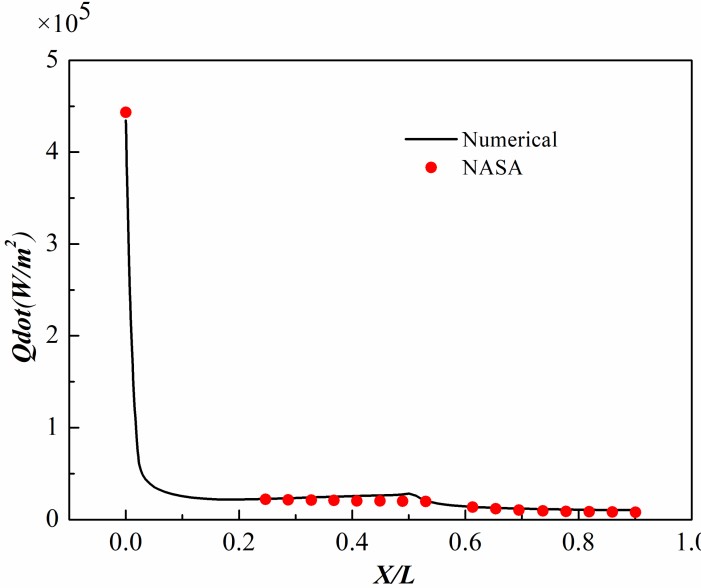

**Figure 7.** Comparison between blunt biconical numerical results and NASA results.

Here, we focus on the heat flow analysis in the stagnation area, so the nose section of the waverider was selected for heat flow analysis. The length of configuration was 100 mm. The computational conditions were Mach number 6, flight altitude 25 km, and turbulence model k-ε-Rt model. The wall temperature was 300 K.

The heat protection effects of two different blunt shapes near the stagnation point heat flow are compared. The heat flux distribution near the stagnation point is important. In order to accurately simulate the heat flow distribution in the stagnation region, the grid near the stagnation point should be densified. The local schematic diagram of the shape grid and the stagnation point heat flow grid is shown in Figure 8.

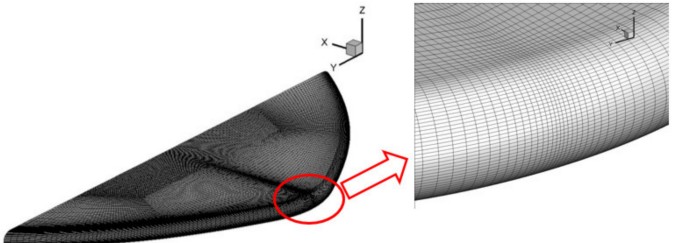

**Figure 8.** Diagram of the stagnation point heat flow mesh.

In order to further verify the accuracy of numerical results, the grid convergence test was carried out. Here, five sets of grids with the same topology but different grid distribution parameters were used for the test. The authors of [29] point out that the near-wall size has a great influence on the heat flow results, and therefore the comparison of different normal grid scale distributions was selected near the wall. In the test calculation example, the blunt radius of 2 mm was taken as an example for analysis.

Computational condition: advection splitting method (AUSM+), implicit time-marching, and a k-$\varepsilon$-Rt model solver. The computed results are shown in Table 2, where D represents the blunt diameter, the angle of attack is 0 degrees, and Qmax represents the maximum heat flow. It can be seen from the Table 2 that when the near-wall size is large (D/2 $\times$ $10^{-2}$ mm), the maximum heat flow does not converge. With the densification of the near-wall size (less than or equal to D/2 $\times$ $10^{-3}$ mm) and the densification of the object surface grid, the numerical deviation of the maximum heat flow did not exceed 4.2%. It can be considered that the numerical results have high reliability. The following numerical calculations were based on the fourth set (D $\times$ $10^{-5}$ mm) of grid parameters.

**Table 2.** The results of grid convergence.

| Number | Near-Wall Size/mm | Qmax/(W/m$^2$) | Grid Numbers |
|:------:|:-----------------:|:--------------:|:------------:|
| 1 | $D/2 \times 10^{-2}$ | $3.57 \times 10^6$ | $1.88 \times 10^6$ |
| 2 | $D/2 \times 10^{-3}$ | $3.97 \times 10^6$ | $1.88 \times 10^6$ |
| 3 | $D/2 \times 10^{-4}$ | $4.04 \times 10^6$ | $1.88 \times 10^6$ |
| 4 | $D \times 10^{-5}$ | $4.15 \times 10^6$ | $1.88 \times 10^6$ |
| 5 | $D \times 10^{-5}$ | $4.12 \times 10^6$ | $3.35 \times 10^6$ |

## 4. Aerothermal Characteristics and Aerodynamic Performance of Blunt Waverider

### 4.1. Flow-Field Comparison

Pressure contour under the symmetry plane of the waverider is shown in Figure 9. The flight conditions were Mach number 6, angle of attack of 0 degrees, and the model diameter/height was 10 mm. As can be seen from Figure 10, the pressure distribution between the two blunt profiles was similar. The difference is the high-pressure region of the non-uniform blunt profile was larger than that of the circular blunt profile.

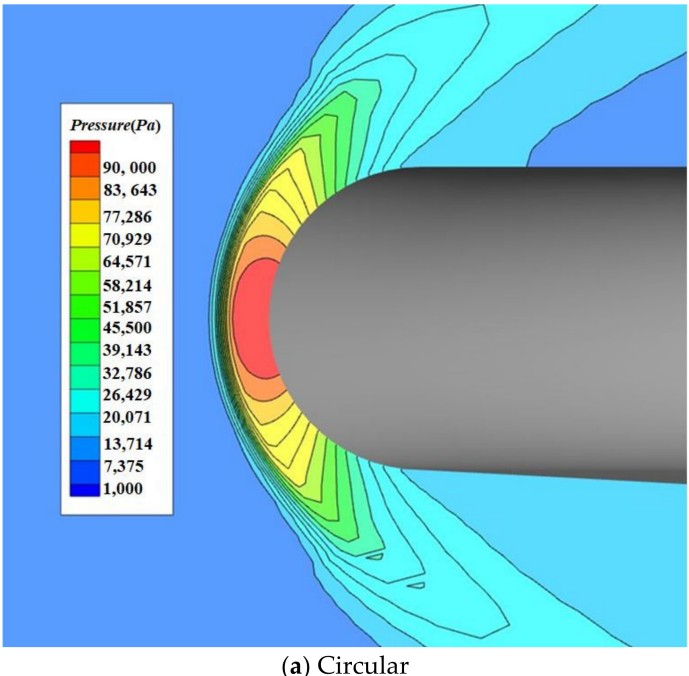

(**a**) Circular

**Figure 9.** *Cont.*

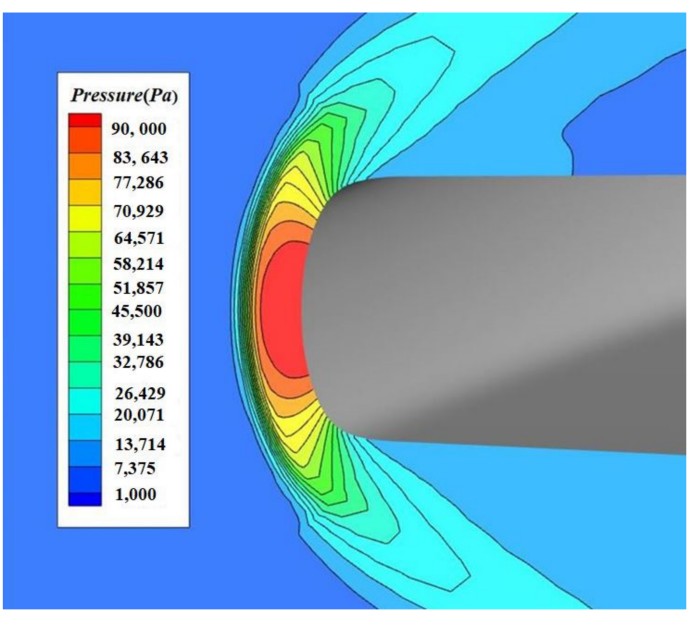

(**b**) Non-uniform

**Figure 9.** Pressure contour under the symmetry plane of the waverider (diameter/height: 20 mm, units: Pa).

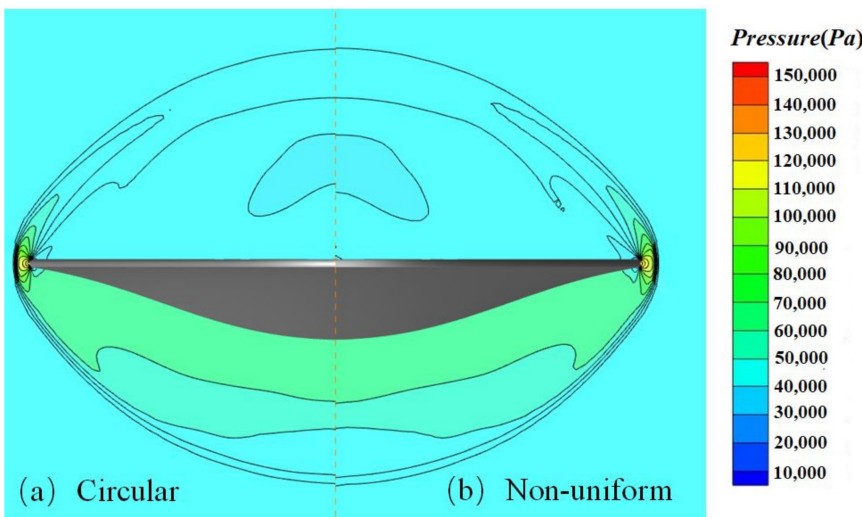

**Figure 10.** Outlet pressure contour of different blunt profiles (diameter/height: 20 mm, units: Pa).

The waverider can be used as the compression surface of the intake port of an aspirated engine, and the quality of its outlet flow field is also a very important index. Outlet pressure contour of different blunt profiles is shown in Figure 10. The outlet pressure contour of the two different blunt profiles had little difference. In other words, the non-uniform blunt profile had little effect on the performance of the waverider.

### 4.2. Effects of Angle of Attack

Heat flux distribution of the two blunt profiles is shown in Figure 11. The diameter/height of the blunt profile of the waverider was 4 mm and the flight angle of attack was 0 degrees. The heat flux of the non-uniform blunt profile was lower than that of the circular blunt. The heat flux distribution near the stagnation point of circular arc passivation was elliptical. The heat flux distribution in the vicinity of the stagnation point of the non-uniform blunt was irregular compared with the circular blunt profile near the stagnation point, whereby the non-uniform blunt profile had lower heat flux near the stagnation point.

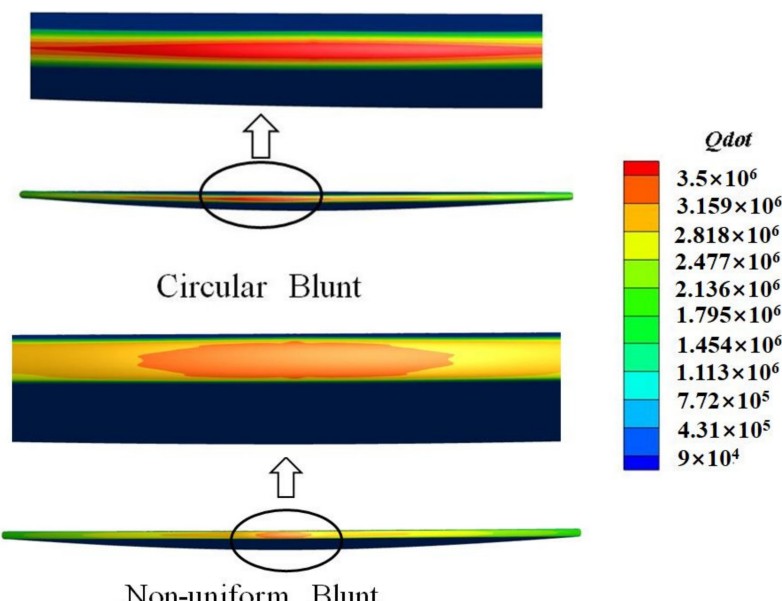

**Figure 11.** Heat flux distribution of two blunt profiles (diameter/height: 4 mm, angle of attack: 0 degrees, units: w/m$^2$).

The peak heat flux of different blunt radii is shown in Figure 12. The abscissa ordinate is the angle of attack, and longitudinal ordinate is the maximum heat flow. Non-uniform shape-H4 indicates that the blunt shape has a non-uniform shape, where H4 indicates that the blunt height is 4 mm. Circular shape-D4 indicates that the blunt shape is circular, where D4 indicates that the dimeter is 4 mm.

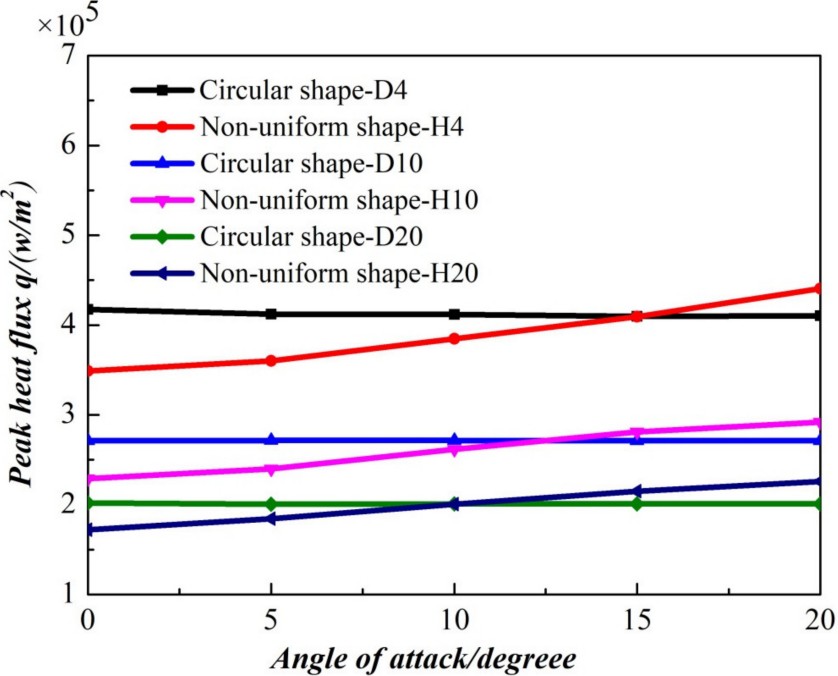

**Figure 12.** Peak heat flux of different blunt radii.

Under the 0-degree angle of attack, the peak heat flux of the non-uniform blunt decreased with the increase of blunt diameter/height compared with that of circular blunt. The reason is that the heat flow at the stagnation point is related to the vertical and horizontal curvature radius of the stagnation point. With the increase of the radius, the equivalent curvature radius between non-uniform blunt and the circular blunt gradually

decreased. When the blunt diameter/height of the blunt waverider was 4 mm, the peak heat flux of the non-uniform blunt profile reduced by about 17% compared to that of the circular blunt profile. When the blunt diameter/height of the blunt waverider was 20 mm, the peak heat flux of the non-uniform blunt profile reduced by about 15% compared to that of the circular blunt profile. With the increase of blunt diameter/height, the range of the attack angle of the non-uniform blunt that can effectively reduce the peak heat flow gradually became smaller, but it can meet the real engineering requirements when the blunt height is 20 mm. When the blunt diameter/height was 4 mm, the peak heat flux of the non-uniform blunt profile was smaller than that of the circular blunt profile within a 15-degree angle of attack. With the increase of the attack angle over 15 degrees, the maximum heat flow of the non-uniform blunt was greater than that of the circular blunt. The reason is that with the increase of the angle of attack, the equivalent curvature radius of the stagnation point of the non-uniform blunt waverider decreased. When the blunt diameter/height was 20 mm, the maximum heat flow of the non-uniform shape was smaller than that of the circular blunt shape in the range of the 10-degree attack angle.

Heat flux distribution along the center line of blunt profiles is shown in Figure 13. Abscissa is the coordinate of the blunt profile, H is the height (4 mm) of the blunt profile, longitudinal is the heat flux, and $Q_0$ is the peak heat flux (4,174,820 W/m$^2$) of the circular blunt profile. Considering that the heat flux distribution curves of the circular blunt deviated as a whole with the increase of the attack angle, and the peak heat flow remained unchanged, therefore, the heat flux distribution at the 0-degree attack angle was selected for analysis. D4-0° indicates that the diameter of the circular blunt profile was 4 mm, and the angle of attack was 0 degrees. H4-0° indicates that the height of the non-uniform blunt profile was 4 mm and the angle of attack was 0 degrees.

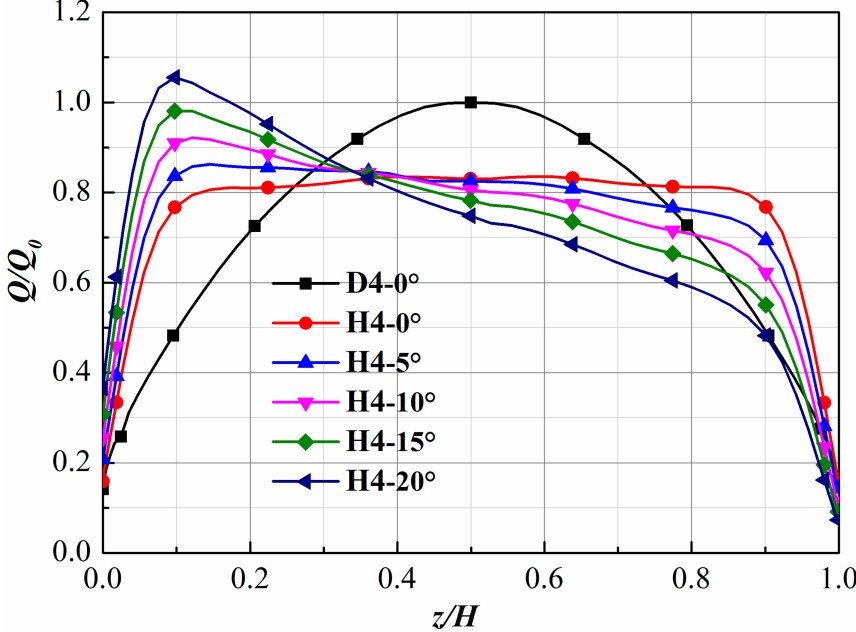

**Figure 13.** Heat flux distribution along the center line of the blunt waverider.

With the increase of the distance from the stagnation point, the heat flux of the circular blunt profile rapidly decreased. The heat flux near the stagnation point of the non-uniform blunt profile at the 0-degree angle of attack showed little change. With a further increase of the distance, the heat flow began to sharply decrease. The above heat flux distribution is caused by the non-uniform blunt curvature radius. With the increase of the angle of attack, the non-uniform blunt heat flow changed from a uniform distribution near the stagnation point to a single peak, and the peak heat value increased. The reason is that the uniform

blunt profile was designed at the 0-degree angle of attack, and the radius of curvature was small far away from the stagnation point.

In order to investigate the three-dimensional heat flux distribution of the blunt waverider, the spanwise heat flux distribution, including peak heat flux of the blunt waverider, was analyzed. The spanwise distribution curves, including peak heat flux, are shown in Figure 14. The abscissa is the coordinate of the blunt waverider, W is half the width of the blunt waverider, longitudinal is the heat flux, and Q0 is the peak heat flux (4,174,820 W/m$^2$) of the circular blunt waverider. Considering that the spanwise distribution of the circular waverider including the peak heat flux was the same under different attack angles, the blunt waverider with a diameter of 4 mm under the 0° attack angle was selected for analysis. D4-0° indicates that the diameter of the circular blunt profile was 4 mm, and the angle of attack was 0 degrees. H4-0° indicates that the height of the non-uniform blunt profile was 4 mm and the angle of attack was 0 degrees.

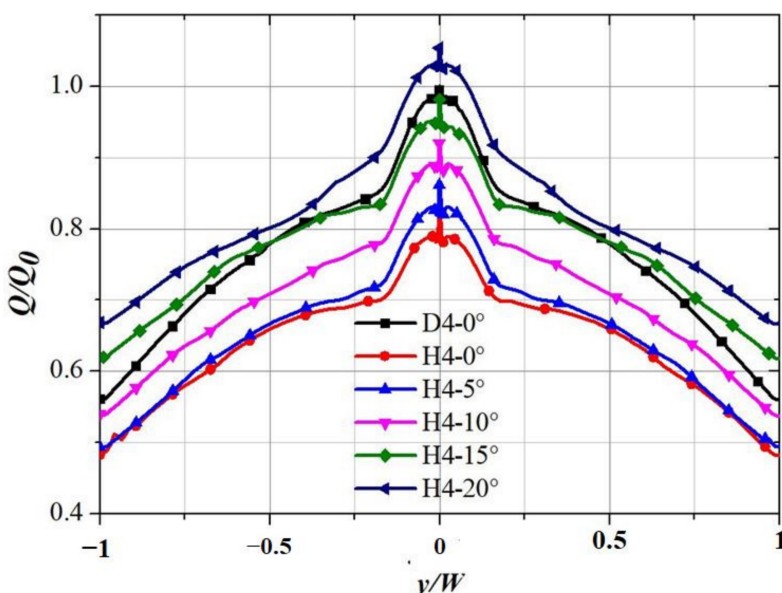

**Figure 14.** Spanwise distribution curves, including peak heat flux.

It can be seen from Figure 14 that the spanwise heat flux, including peak heat flux, under different angles of attack was similar. The spanwise heat flux distribution curves of the non-uniform blunt under different attack angles were not parallel, because the curvature of the non-uniform curves containing peak heat flux was different. Non-uniform blunt spanwise heat flux in the range of the 10° attack angle was lower than that of the circular blunt under the 0° attack angle. At the 15-degree flight attack angle, the heat flux of the non-uniform blunt was lower than that of the circular blunt in the span width range of −0.5 to 0.5, and that of other regions was higher than that of circular blunt. At the 20-degree flight angle of attack, the non-uniform blunt spanwise heat flux was higher than the circular blunt heat flux. The above spanwise heat flux distribution is caused by the interaction of the non-uniform blunt curvature radius and the waverider edge curvature radius.

A comparison of the drag of different blunt profiles is shown in Figure 15. The flight Mach number was 6, the flight altitude was 25 km, and the angle of attack was 2 degrees (the maximum lift-to-drag ratio of the sharp waverider was at the angle of attack of 2 degrees). Circular indicates the circular blunt waverider, and non-uniform indicates the non-uniform blunt profile.

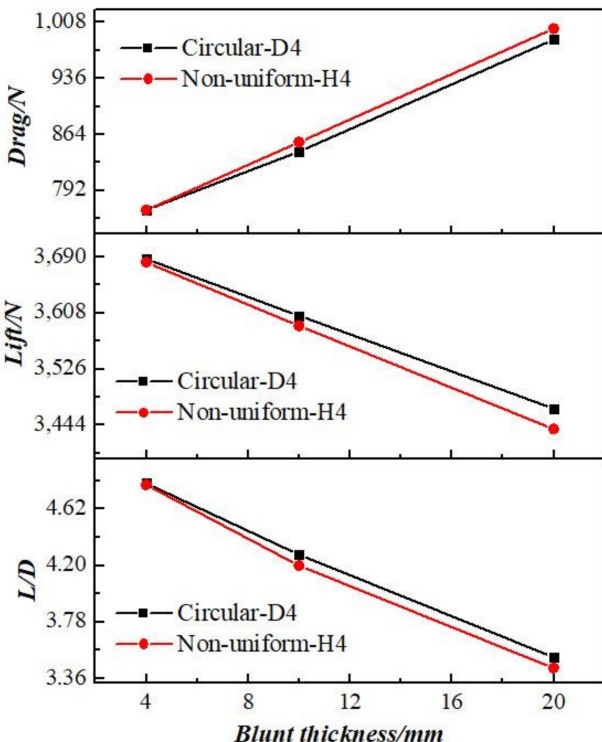

**Figure 15.** Aerodynamic performance force of the two blunt profiles' thickness (angle of attack: 2 degrees).

As can be seen from Figure 15, with the increase of blunt diameter/height, the drag of the blunt waverider increased. Under the same blunt diameter/height, the high-pressure area of the non-uniform blunt was slightly larger than that of the circular blunt, and therefore the drag of the non-uniform shape was slightly larger than that of the circular shape. With the increase of the blunt diameter/height, the difference gradually increased. When the blunt diameter/height was 20 mm, the drag difference between the two blunt shapes was only 1.4%.

Under the same blunt thickness, the lift of the non-uniform blunt profile was smaller than that of the circular blunt profile. The reason is that the high-pressure leakage on the upper surface of the non-uniform blunt was higher than that of the circular blunt. As can be seen from Figure 15, there was little difference in the lift between the two blunt profiles, the reason being that the difference between the normal projection areas of the two blunt profiles was small. With the increase of blunt thickness, the difference between the normal projection areas of the two blunt profiles gradually increased, and the lift gradually decreased.

As can be seen from Figure 15, under the same blunt height, the lift-to-drag ratio of the non-uniform blunt profile was smaller than that of the circular blunt profile. With the increase of the blunt height, the difference in the lift-to-drag ratio between the two blunt shapes gradually increased. When the blunt thickness was 20 mm, the lift-to-drag ratio of the non-uniform blunt reduced by 2.3% compared to that of the circular blunt.

The aerodynamic performance of the different blunt profiles with the angle of attack is shown in Figure 16. The blunt diameter/height of the waverider was 20 mm. Circular-D20 indicates that the circular blunt diameter was 20 mm, and non-uniform-H20 indicates that the non-uniform blunt height was 20 mm. As can be seen from Figure 16, there was little difference between the aerodynamic performance force performances of the two different blunt configurations with the angle of attack. Considering that the flow direction projection area and the normal projection area of the two blunt profiles had little difference, therefore, the difference in drag and lift between the two blunt profiles was very small. The lift-to-drag ratio of the two blunt profiles reached the maximum value at a 5-degree angle

of attack. The lift-to-drag ratio difference between the two blunt profiles also reached a maximum of 2.5%. The difference between the lift-to-drag ratios of the two profiles was slightly large within the range of the 10-degree attack angle, which may be caused by the numerical error of the calculation method.

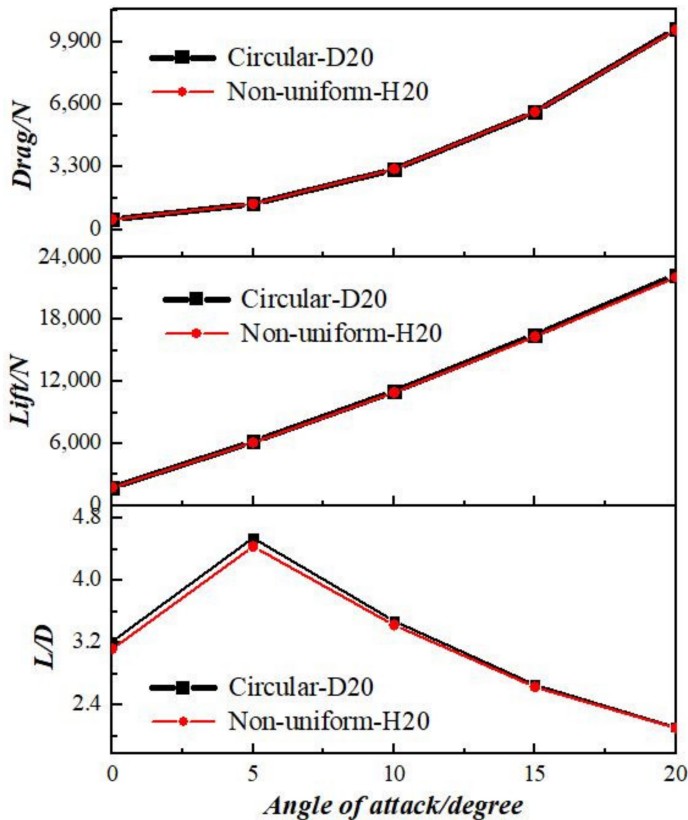

**Figure 16.** Aerodynamic performance of different blunt profiles with angle of attack (diameter/height: 20 mm).

### 4.3. Effects of Flight Altitudes

Under the conditions of a wall temperature of 300 K, angle of attack of 0°, and Mach number of 6, several cases with different flight altitudes ranging from 25 km to 40 km have been studied to assess the aerothermal characteristic performance. A comparison of the peak heat flux for various Mach numbers is shown in Figure 17. The abscissa is the blunt height/diameter, and the longitudinal coordinates show peak heat flux. Circular-H25 represents the circular blunt waverider, where the flight altitude was 25 km. With the increase of flight altitudes, the incoming flow density became smaller, and the peak heat flux was proportional to the 1/2 power of the density, so the peak heat flux decreased. Under the same blunt radius/height, the peak heat flux of the non-uniform blunt profile decreased by about 17% compared to the circular blunt, which was due to the combined action of the non-uniform blunt curvature radius and the edge blunt curvature radius of the waverider. Therefore, it is feasible and reliable to reduce the peak heat flux of the blunt waverider by using a non-uniform profile.

The aerodynamic performance of different blunt profiles with flight altitudes is shown in Figure 18. Circular-D20 indicates that the circular blunt diameter was 20 mm, and non-uniform-H20 indicates that the non-uniform blunt height was 20 mm. Flight conditions were a Mach number of 6 and the flight angle of attack was 0 degrees. There was little difference in the aerodynamic force performance between the two blunt profiles at flight altitudes of 25 to 40 km. With the increase of the height, the incoming flow density became smaller, and therefore the lift and drag of the waverider of the two blunt profiles decreased. The maximum difference between the drag and lift of the two blunt shapes did not exceed

3% at a flight altitude of 25 km to 35 km, and did not exceed 5% at a flight altitude of 40 km. When the flight height of the blunt waverider was 40 km, the aerodynamic values of the two blunt profiles were small, and the drag difference was about 3 N, but the ratio difference between the two blunt profiles was slightly larger (4.5%).

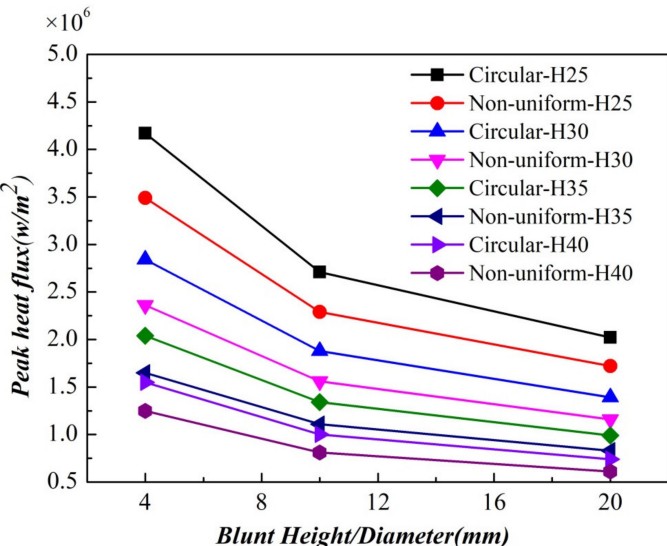

**Figure 17.** Comparison of peak heat flux for various flight altitudes.

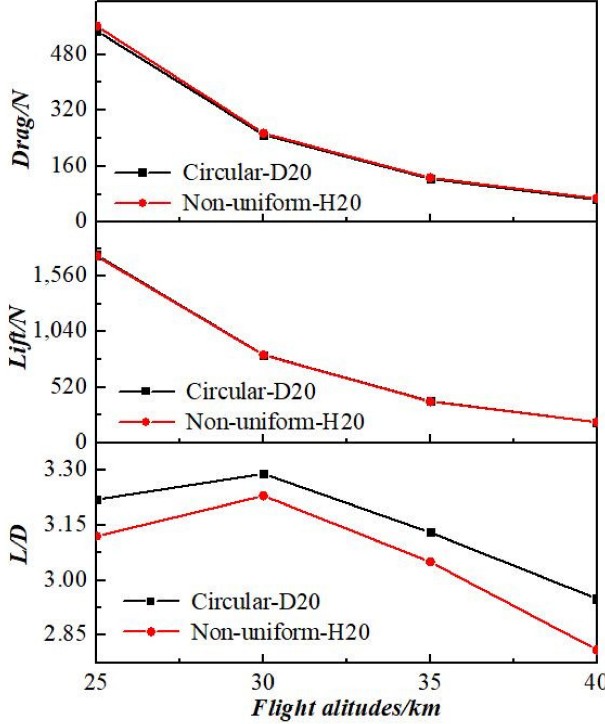

**Figure 18.** Aerodynamic performance of different blunt profiles with various flight altitudes (diameter/height: 20 mm).

### 4.4. Effects of Mach Numbers

Under the conditions of a wall temperature of 300 K, flight altitude of 25 km, and an angle of attack of 0°, several cases with different flight Mach numbers ranging from 2 to 10 have been studied for the aerothermal characteristic performance. A comparison of peak heat flux for various Mach numbers is shown in Table 3. With the increase of Mach numbers, the peak heat flux of the two blunt profiles gradually increased. Under the same

blunt diameter/height, the peak heat flux of the non-uniform blunt profile reduced by about 17% compared to the circular blunt, which was due to the combined action of the non-uniform blunt curvature radius and the edge blunt curvature radius of the waverider. The waverider with a uniform blunt profile can efficiently reduce the value of peak heat flux in a wide range of Mach numbers.

**Table 3.** Comparison of peak heat flux for various Mach numbers (unit: $W/m^2$).

| Diameter/Height | Blunt Profile | Mach Numbers | | | | |
|---|---|---|---|---|---|---|
| | | 2 | 4 | 6 | 8 | 10 |
| 4 mm | Circular | $0.83 \times 10^{+5}$ | $1.08 \times 10^{+6}$ | $4.17 \times 10^{+6}$ | $1.03 \times 10^{+7}$ | $2.06 \times 10^{+7}$ |
| | Non-uniform | $0.70 \times 10^{+5}$ | $0.89 \times 10^{+6}$ | $3.49 \times 10^{+6}$ | $0.87 \times 10^{+7}$ | $1.74 \times 10^{+7}$ |
| | Reduced by | +16% | +18% | +17% | +16% | +16% |
| 10 mm | Circular | $0.52 \times 10^{+5}$ | $0.71 \times 10^{+6}$ | $2.71 \times 10^{+6}$ | $0.69 \times 10^{+7}$ | $1.36 \times 10^{+7}$ |
| | Non-uniform | $0.43 \times 10^{+5}$ | $0.59 \times 10^{+6}$ | $2.29 \times 10^{+6}$ | $0.57 \times 10^{+7}$ | $1.13 \times 10^{+7}$ |
| | Reduced by | +17% | +17% | +16% | +17% | +17% |
| 20 mm | Circular | $0.37 \times 10^{+5}$ | $0.52 \times 10^{+6}$ | $2.02 \times 10^{+6}$ | $0.50 \times 10^{+7}$ | $1.01 \times 10^{+7}$ |
| | Non-uniform | $0.31 \times 10^{+5}$ | $0.43 \times 10^{+6}$ | $1.72 \times 10^{+6}$ | $0.42 \times 10^{+7}$ | $0.84 \times 10^{+7}$ |
| | Reduced by | +16% | +16% | +15% | +16% | +17% |

The aerodynamic performance of different blunt profiles with Mach numbers is shown in Figure 19. Circular-D20 indicates that the circular blunt diameter was 20 mm, and non-uniform-H20 indicates that the non-uniform blunt height was 20 mm. Flight conditions were a Mach number of 6, flight altitude of 25 km, and a flight angle of attack of 0 degrees. As can be seen from Figure 19, there was little difference in the aerodynamic force performance between the two blunt profiles. With the increase of Mach numbers, the incoming flow velocity became larger, and therefore the drag and lift of the waverider of the blunt profile gradually increased, and the difference was no more than 3.6% between the two blunt profiles. The lift-to-drag ratio of the two blunt profiles gradually decreased, and the difference was no more than 4.5%.

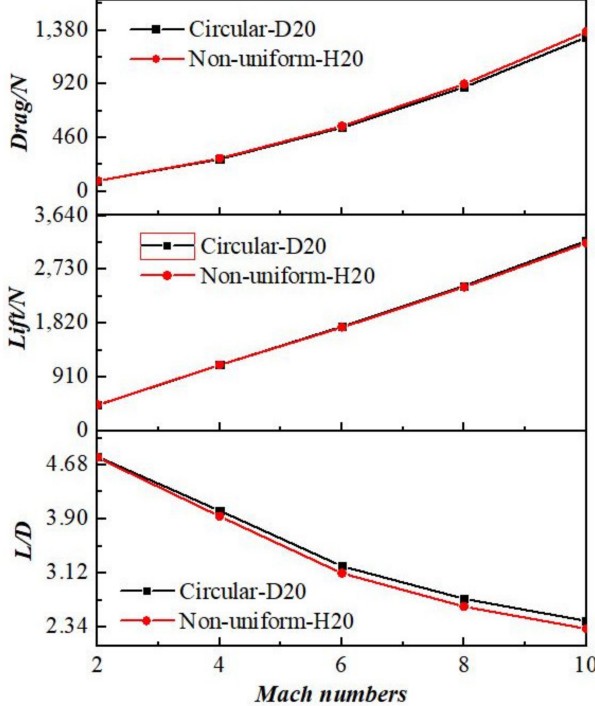

**Figure 19.** Aerodynamic performance of different blunt profiles with various Mach numbers (diameter/height: 20 mm).

## 5. Conclusions

In this paper, the influence of the non-uniform blunt profile on the aerothermal characteristics and aerodynamic performance of the waverider was studied by numerical simulation. The following conclusions were obtained:

The aerothermal performance of the non-uniform blunt waverider under different flight angles of attack, flight altitudes, and flight Mach numbers has been studied. The results showed that the peak heat flux of the waverider with non-uniform blunt reduced by about 17% compared with that with circular blunt under a 0-degree angle of attack, with different flight altitudes and Mach numbers. With the increase of the blunt diameter/height, the attack angle of the non-uniform blunt peak heat flux decreased compared to the circular blunt peak heat flux.

The aerodynamic performance of the non-uniform blunt waverider under different flight angles of attack, flight altitudes, and flight Mach numbers has been studied. The results showed that the non-uniform blunt shape had little effect on the drag of the waverider with the increase of the blunt radius. Even when the blunt height/diameter was 20 mm, the aerodynamic difference between the two different blunt profiles did not exceed 3% within a 15-degree angle of attack, Mach 2-10, and flight altitude of 15–35 km.

Compared with the aerodynamic performance of heat flux and the aerodynamic performance of the circular blunt, the non-uniform blunt could effectively reduce the heat flow and had little impact on the aerodynamic performance. Therefore, it is feasible and reliable to be applied to hypersonic waverider vehicles by using a non-uniform profile.

**Author Contributions:** Methodology, W.W.; Validation, Y.X.; Formal analysis, G.L.; Investigation, H.X.; Writing—review and editing, Z.Q.; Supervision, K.C.; Funding acquisition, G.L. All authors have read and agreed to the published version of the manuscript.

**Funding:** This work was funded by the National Natural Science Foundation of China (Grant No. 12002347) and the Henan Provincial Department of Science and Technology Research Project (Grant No. 222102220069).

**Acknowledgments:** This work was supported by the National Natural Science Foundation of China (Grant No. 12002347) and the Henan Provincial Department of Science and Technology Research Project (Grant No. 222102220069).

**Conflicts of Interest:** The authors declare no conflict of interest.

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
