# Peer review of "Influence of Non-Uniform Bluntness on Aerodynamic Performance and Aerothermal Characteristics of Waverider"

_aerospace, doi:10.3390/aerospace10030205_

Round 1

Reviewer 1 Report

Referee review

This paper presents a numerical study on the influence of the bluntness on the aerodynamic performance of a wave rider. The originality of this study concerns the comparison between a circular blunt geometry and a non-uniform blunt geometry.

The simulations were performed with several geometries, Mach numbers and flight altitudes.  The results concern aerodynamic coefficients and heat fluxes. Interesting information for the scientific community is presented in this paper but needs to be completed to be accepted for publication. Indeed the results presented are partial compared to the study which is announced at the beginning of the manuscript.

Major revision

1.       A more detailed state of the art comparing this study with other studies in the literature should be added to place this study in a more general context. Reporting in the introduction by citing 20 references without any explanation makes the introduction and understanding of the problem difficult for the reader.

2.       For each test case, the calculation conditions are not clear. Especially the wall conditions which should have an importance on the heat flow results. Some clarifications should be added.

3.       If I understand correctly, the wall temperature is 300 K?  How do you justify this condition, especially for the Mach 10 test case?  This could lead to underestimated results? What about viscous layers for high Mach numbers? How do you take this into account?

4.       Related to previous question, it should be useful to present the surface distribution temperature for the different test cases studied before results of heat fluxes.

5.       The results in table 3 should be easier to read with a figure.

6.       The authors state that "the numerical error of the numerical calculation method" could lead to a great aerodynamic performance at 40 km. This statement should be explained and clarified.

7.       It would be interesting to present, with figures, the influence of Mach effects for all the studied altitudes in order to evidence the Mach and density effects. In the paper only one test case is presented.

Minor revision

1.       Figure 11 is not clear because it is too small.

2.       A diagram of the waverider geometry must be added. With the specific characteristics.

3.       The quality of figures has to be improved

4.       Some references have to be re written in the good format for the journal

5.       Check that all table titles begin with a capital letter

Author Response

List of Responses

Dear Editors and Reviewers:

Thank you for your letter and comments from the reviewers about our manuscript ID aerospace-2155877. Those comments are all valuable and very helpful for revising and improving our paper, as well as the important guiding significance to our researchers. We have carefully checked the manuscript and revised it according to the reviewer’s comments. We submit here the revised manuscript in which the revised portions are marked in red color. The responds to the reviewers are listed below.Major revision

  • Comments 1.

A more detailed state of the art comparing this study with other studies in the literature should be added to place this study in a more general context. Reporting in the introduction by citing 20 references without any explanation makes the introduction and understanding of the problem difficult for the reader.

Reply 1:

Thanks for your perceptive comments and kind suggestions. The literature is summarized, analyzed and studied. See the manuscript for relevant contents.

  • Comments 2.

 For each test case, the calculation conditions are not clear. Especially the wall conditions which should have an importance on the heat flow results. Some clarifications should be added.
 Reply 2:

Thanks for your kind suggestion. It has been modified according to your suggestion. See the manuscript for the modified part. See the manuscript for more details.

  • Comments 3.

If I understand correctly, the wall temperature is 300 K?  How do you justify this condition, especially for the Mach 10 test case?  This could lead to underestimated results? What about viscous layers for high Mach numbers? How do you take this into account?

Reply 3:

Thanks for your perceptive comments and kind suggestions. The wall temperature is related to many factors, including flight Mach number, flight trajectory, aircraft structure and material, etc. The wall temperature is usually higher than 300K at Mach 10, but the accurate temperature is difficult to determine. The purpose of this paper is to compare and analyze the influence of two kinds of blunt profiles on the maximum heat flow. Therefore, the cold wall condition (wall temperature 300k) is uniformly selected for calculation. In addition, we have also made a detailed analysis of the effect of different wall temperatures on the heat flow, as shown in Table 1 . The results show that the heat flow value will indeed decrease after the wall temperature increases, but the optimized leading edge or the non-uniform blunt leading edge can be reduced by about 20% compared with the circular blunt. Therefore, 300K cold wall is selected for analysis.

Table 1 Comparison of peak heat flux at different wall temperatures

Wall Temperature(K)

Baseline(W/m2)

Optimal(W/m2)

Reduced by

300

4.13 E+06

3.31 E+06

+20%

600

3.47 E+06

2.79 E+06

+20%

900

2.80 E+06

2.23 E+06

+20%

1200

2.11 E+06

1.69 E+06

+20%

1500

1.43 E+06

1.14 E+06

+20%

1800

7.37 E+05

5.85 E+05

+21%

With regard to the effect of viscosity at high Mach number, the three-dimensional NS equations in the ideal gas state is used to solve the problem. The effect of high temperature chemical reaction is not considered at present, and k-Ɛ-Rt model is selected as the turbulence model, in order to simulate the boundary layer more accurately, the normal direction of the wall is densified.

  • Comments 4.

 Related to previous question, it should be useful to present the surface distribution temperature for the different test cases studied before results of heat fluxes.

Reply 4:

Thanks for your perceptive comments and kind suggestions. This paper focuses on the analysis of the influence of heat flow. The wall is uniformly given to the cold wall isothermal wall. When the wall temperature rises, that is, the hot wall heat flow. At the same temperature, the hot wall heat flow is proportional to the cold wall heat flow. Therefore, the author believes that it is meaningful to use the cold wall condition to compare and analyze the heat flow. This paper studies the heat reduction effect of the peak heat flow of the non-uniform blunt shape, analyzes the stagnation point leading edge profile and the spanwise heat flow distribution. Figure 11 shows the heat flow distribution of the three-dimensional surface.

  • Comments 5.

  The results in table 3 should be easier to read with a figure.

Reply 5:

Thanks for your perceptive comments and kind suggestions. According to your suggestion, change Table 3 to Figure 17.

  • Comments 6.

 The authors state that "the numerical error of the numerical calculation method" could lead to a great aerodynamic performance at 40 km. This statement should be explained and clarified.

Reply 6:

Thanks for your perceptive comments and kind suggestions. When the flight height of the blunt waverider is 40 km, the aerodynamic values of the two are small, and the drag difference is about 3 N, but the ratio difference between the two blunt profiles is slight larger (4.5%).

  • Comments 7.

 It would be interesting to present, with figures, the influence of Mach effects for all the studied altitudes in order to evidence the Mach and density effects. In the paper only one test case is presented.

Reply 7:

Thanks for your perceptive comments and kind suggestions. Pressure contours of two configurations at different Mach numbers are shown in Figure 1. The pressure contours of the two configurations at different Mach numbers have little difference, so the aerodynamic performance of the two configurations has little difference.

It can be seen from Figure 1 that the two shape mechanisms are the same, limited to the length of the paper, and the aerodynamic performance analysis of different flight heights is carried out using single-factor variables, taking Mach 6 as the typical flight condition.

(a) Pressure contours of two different configurations at Mach 2

(b) Pressure contours of two different configurations at Mach 4

(c) Pressure contours of two different configurations at Mach 6

(d) Pressure contours of two different configurations at Mach 8

(e) Pressure contours of two different configurations at Mach 10

Figure 1 Pressure contours of two configurations at different Mach numbers

Minor revision

  • Comments 1.

 Figure 11 is not clear because it is too small.

Reply 1:

Thanks for your perceptive comments and kind suggestions. It has been modified according to your suggestion. See the manuscript for the modified part.

  • Comments 2.

A diagram of the waverider geometry must be added. With the specific characteristics

Reply 2:

Thanks for your perceptive comments and kind suggestions. A diagram of the waverider geometry has added. In the model introduction part, the aerodynamic characteristics of the waverider are introduced

  • Comments 3.

The quality of figures has to be improved.

Reply 3:

Thanks for your perceptive comments and kind suggestions. It has been modified according to your suggestion. See the manuscript for the modified part.

  • Comments 4.

Some references have to be re written in the good format for the journal.

Reply 4:

Thanks for your perceptive comments and kind suggestions. It has been modified according to your suggestion. See the manuscript for the modified part.

  • Comments 5.

Check that all table titles begin with a capital letter.

Reply 5:

Thanks for your perceptive comments and kind suggestions. All table titles begin with a capital letter.

If you have any other suggestions for this article, please give me the opportunity to modify it.

Reviewer 2 Report

Dear Authors,

The manuscript talks about the effect of leading edge choice in a hypersonic wave-rider - circular and non-uniform. It is a good paper and can be published after a minor revision as below

1) Can you describe why or how the non-uniform profile was designed? Describe more from the literature.

2) Can you add boundary layer plots for both profiles?

3) Please enlarge the images for clarity.

4) Please describe the solver used, numerical discretization and convergence technique implemented.

Author Response

List of Responses

Dear Editors and Reviewers:

Thank you for your letter and comments from the reviewers about our manuscript ID aerospace-2155877. Those comments are all valuable and very helpful for revising and improving our paper, as well as the important guiding significance to our researchers. We have carefully checked the manuscript and revised it according to the reviewer’s comments. We submit here the revised manuscript in which the revised portion are marked in red color. The responds to the reviewers are listed below.

  • Comments 1.

Can you describe why or how the non-uniform profile was designed? Describe more from the literature.

Reply 1:

Thanks for your perceptive comments and kind suggestions. For circular blunt produce point ablation at the stagnation point, to reduce the peak heat-flux of circular blunt, the Mini-Max optimization model is introduced for aerothermodynamics optimization. The surface heat-flux is obtained by resolving Navier-Stokes equations, and only the frozen flow is considered. The computational fluid dynamics (CFD) based Genetic Algorithm is used as the optimizer. A novel 2-D profile of leading-edge is obtained and the peak heat-flux is significantly reduced.

Design process:
Figure 1 shows the procedure of the optimization, including three steps as follows:
1) Parametric design of the leading-edge.
2) Solution of the Mini-Max optimization problem.
3) Robustness analysis of the optimal leading-edge.

Figure 1 Schematic of Optimization Procedure

Figure 2 shows the parametric design of leading-edge by using subsection cubic B-spline interpolation. Seven control points, P1~P4 and P1’~P3’ are used for the B-spline, as shown in Figure 2. Here, the coordinate values of P1 and P1’ are determined by the thickness of the leading-edge. As the leading-edge is up-down symmetry, the x coordinate value of P4 is determined to be zero, and P2’, P3’ are set to be the symmetry point relative to the Y-coordinate-axis of P2 and P3, respectively. And thus the profile of the leading-edge is finally parameterized by five design variables only, as shown in Table1 .

Table 1 Design variables’ description

Design variables

ranges

description

V1

[-1.5,-0.5]

x coordinate value of P2

V2

[1.8,2.5]

y coordinate value of P2

V3

[-3.0,-1.5]

x coordinate value of P3

V4

[0.5,1.8]

y coordinate value of P3

V5

[-3.0,-0.5]

x coordinate value of P4

Figure 2 Schematic of the parametric leading-edge (7 control-points and 5 design-variables

 were involved)

The genetic algorithm searches for the optimal value by scattering points in the whole region, and its optimization iteration process mimics the inheritance and mutation functions in the biological genetic process. Here, the population size is set as 10, the double vector coding is adopted, and the number of elitists in each generation is 2, the Gaussian mutation function is adopted in the genetic process, and the crossover probability is set as 0.8, and the population evolution algebra is 30. After 300 optimization iterations, the objective function gradually approaches the global optimal value (as shown in Figure 3), and the leading edge profile also gradually approaches the optimal shape (as shown in Figure 4). Figure 4 shows the optimal leading edge and arc leading edge shapes in the first, seventeenth and thirtieth generations of population, where the shape at i=298 in the thirtieth generation is the final optimized shape.

Figure 3 Peak heat-flux history during the optimization process

Figure 4 Optimal shapes during the optimization process.

Heat-Flux distribution comparison between Baseline(dashed line) and Optimal leading-edge(solid line) is shown in Figure 5. The heat-flux of the optimized 2-D profile distributed relative uniform than that of the circular profile in the vicinity of the stagnation point. The peak heat-flux is decreased about 20% and the heat-flux distribution is little changed in the vicinity of stagnation point.

Figure 5 Heat-Flux distribution comparison between Baseline(dashed line) and Optimal leading-edge(solid line)

  • Comments 2.

Can you add boundary layer plots for both profiles?
 Reply 2:

Thanks for your kind suggestion, the boundary layers of two different blunt shapes are shown in Figure 6, the difference of boundary layer distribution between the two blunt profiles is small.

Figure 6 Comparison of boundary layer distribution of two blunt profiles

  • Comments 3.

Please enlarge the images for clarity.

Reply 3:

Thanks for your perceptive comments and kind suggestions. According to your suggestion, some pictures have been enlarged.

  • Comments 4.

Please describe the solver used, numerical discretization and convergence technique implemented.

Reply 4:

Thanks for your perceptive comments and kind suggestions. Computational condition: advection splitting method, implicit time-marching, and a k-ε-Rt model solver. The maximum first order diffusion is not exceed 0.3.

If you have any other suggestions for this article, please give me the opportunity to modify it.

Round 2

Reviewer 1 Report

I accept the revised paper to be published

Author Response

Thank you very much!
